# Predicting Daily Dry Matter Intake Using Feed Intake of First Two Hours after Feeding in Mid and Late Lactation Dairy Cows with Fed Ration Three Times Per Day

**DOI:** 10.3390/ani11010104

**Published:** 2021-01-06

**Authors:** Shulin Liang, Chaoqun Wu, Wenchao Peng, Jian-Xin Liu, Hui-Zeng Sun

**Affiliations:** Institute of Dairy Science, MoE Key Laboratory of Molecular Animal Nutrition, College of Animal Sciences, Zhejiang University, Hangzhou 310058, China; liangshulin@zju.edu.cn (S.L.); 18845042935wcq@sina.com (C.W.); happypwc@foxmail.com (W.P.); liujx@zju.edu.cn (J.-X.L.)

**Keywords:** dry matter intake, predicted model, dairy cow

## Abstract

**Simple Summary:**

It is difficult to obtain feed intake of dairy cows in experiments since all cows are raised in a free-stall barn in commercial dairy farms nowadays. Therefore, it is necessary to develop a simple, accurate, and reliable feed intake prediction model to replace direct measurement. In this study, we generated a forecasting model to predict daily dry matter intake (DMI) of dairy cows in mid and late lactation based on the feed intake of first 2 h after feeding (DMI-2h). The proposed prediction equation was: DMI (kg/day) = 8.499 + 0.2725 × DMI-2h (kg/day) + 0.2132 × Milk yield (kg/day) + 0.0095 × Body weight (kg/day) (R^2^ = 0.46). Compared with NRC model (2001), our model shows higher accuracy and precision on predicting daily DMI of dairy cows with fed ration three times per day in mid and late lactation period. This prediction model could be used as an alternative approach for researchers who have difficulty in measuring DMI in dairy cows’ experiments.

**Abstract:**

The objective of this study was to evaluate the feasibility of using the dry matter intake of first 2 h after feeding (DMI-2h), body weight (BW), and milk yield to estimate daily DMI in mid and late lactating dairy cows with fed ration three times per day. Our dataset included 2840 individual observations from 76 cows enrolled in two studies, of which 2259 observations served as development dataset (DDS) from 54 cows and 581 observations acted as the validation dataset (VDS) from 22 cows. The descriptive statistics of these variables were 26.0 ± 2.77 kg/day (mean ± standard deviation) of DMI, 14.9 ± 3.68 kg/day of DMI-2h, 35.0 ± 5.48 kg/day of milk yield, and 636 ± 82.6 kg/day of BW in DDS and 23.2 ± 4.72 kg/day of DMI, 12.6 ± 4.08 kg/day of DMI-2h, 30.4 ± 5.85 kg/day of milk yield, and 597 ± 63.7 kg/day of BW in VDS, respectively. A multiple regression analysis was conducted using the REG procedure of SAS to develop the forecasting models for DMI. The proposed prediction equation was: DMI (kg/day) = 8.499 + 0.2725 × DMI-2h (kg/day) + 0.2132 × Milk yield (kg/day) + 0.0095 × BW (kg/day) (R^2^ = 0.46, mean bias = 0 kg/day, RMSPE = 1.26 kg/day). Moreover, when compared with the prediction equation for DMI in Nutrient Requirements of Dairy Cattle (2001) using the independent dataset (VDS), our proposed model shows higher R^2^ (0.22 vs. 0.07) and smaller mean bias (−0.10 vs. 1.52 kg/day) and RMSPE (1.77 vs. 2.34 kg/day). Overall, we constructed a feasible forecasting model with better precision and accuracy in predicting daily DMI of dairy cows in mid and late lactation when fed ration three times per day.

## 1. Introduction

Feed efficiency in dairy industry attracts increasing interests of researchers [1]. Generally, feed efficiency refers to the ratio of produced milk to consumed feed in dairy cows, in which the dry matter intake (DMI) serves as an important indicator and essential component in feed efficiency calculation and precise ration formulation [2]. For cows housed in a tie-stall barn, it is applicable to determine the daily DMI by recording the weight of added and remainder of the ration. However, it has become difficult to measure the daily DMI of individual dairy cows raised in free-stall houses (the major management type in the modern dairy industry) [1]. Therefore, it is necessary to develop an accurate, precise and reliable prediction model of DMI, which will be of practical significance to carry out experimental research on dairy cows.

Actually, many DMI prediction models have been developed in recent decades. These models range from simple models that only include animal factors (e.g., NRC (2001) model) to complex theoretical models embracing animal and food characteristics and environmental influences [3,4,5]. Among them, the NRC (2001) model was the simplest but most classic model since it was developed using a long-term (1988–1998) dataset from 17,087 Holstein cows fed a wide range of diet types [6]. Therefore, NRC model was considered as an accurate and reliable model and frequently cited as a comparing reference model in the development of new models [5]. Nevertheless, the NRC model as well as other complex models were mainly used to estimate the average DMI of dairy cows (e.g., weekly DMI), and may be not suitable for predicting daily DMI. Compared with the weekly DMI, the daily DMI is difficult to predict precisely due to greater variability [7,8]. Therefore, parameters that are more closely related to daily DMI may be required to increase the accuracy of prediction.

Feed intake of dairy cows is closely related to feeding behavior, which includes eating, chewing, and rumination [9], indicating that feeding behavior may be able to predict daily feed intake more accurately. In recent research, Johnston and DeVries emonstrated, using data from multiple studies of high production dairy cows, that meal frequency and total feeding time were strong predictors of daily DMI [10]. However, there are many factors affecting feeding behaviors of dairy cows, such as feed bunk space, feed barrier design, stocking density, ration composition, and so on. Dairy cows are social animals and easily generate dominance hierarchies; if cows are unable to gain access to feed when desired, as in the case of spatial restriction (e.g., insufficient feed bunk space or high stocking density) or temporal feed restriction (e.g., how long feed is available throughout the day), the competition between individuals will increase, which has negative impacts on feeding behavior [11]. Therefore, feeding behavior of dairy cows with different feeding management may be inconsistent, thereby affecting the accuracy of the DMI prediction model. For example, a study based on the NRC model and adding the variable of ruminating time to predict DMI shows that adding ruminating time did not improve the accuracy of predicting DMI [12].

To date, the research using feeding behavior to predict DMI is still limited. One major reason is that the data of feeding behavior is not easily obtained if there is no electronic device or monitor [13]. Some researchers investigated the feeding behavior of cows throughout the day and found that the first 1–2 h after feeding was the peak feeding period, which accounted for the majority of total feed intake [14,15]. In the experiment on dairy cows, it is feasible and easy to record the DMI of short term after feeding (e.g., 2 h) compared with recording the DMI of 24 h or other information of feeding behavior (e.g., eating, chewing, and ruminating time). Thus, it is speculated that DMI of first 2 h after feeding (DMI-2h) will be closely related to total DMI and can act as a variable to accurately predict daily DMI. Moreover, only recording the DMI-2h to predict total DMI under practical experiment would significantly reduce the difficulty in obtaining DMI data for dairy cow researchers.

Therefore, the objective of this study was to investigate the correlation between DMI-2h and total DMI and establish an accurate forecasting model of DMI based on the DMI-2h.

## 2. Materials and Methods

### 2.1. Collection and Preprocessing of DMI Data

Totally, 2700 daily DMI data were collected to generate the development dataset (DDS) for model development by recording feed intake of 54 multi-parity Holstein dairy cows (BW = 621 ± 78.2 kg, DIM = 170 ± 21.7 day) during 50 consecutive days. The 54 cows were selected from two experimental batches, which conducted in the same dairy farm (Zhengxing Animal Industries, Hangzhou, Zhejiang, China) from March to June 2017 and from March to June 2019, respectively. The two batches were under the same experimental design, similar animal conditions (DMI, milk yield, DIM, and parity), and same total mixed ration (TMR) (Appendix A). In total, 660 daily DMI data were obtained from 22 Holstein cows (10 primiparity and 12 multiparity, BW = 577 ± 64.3 kg, DIM = 221 ± 54.0 day) in another 30-day experiment to generate the validation dataset (VDS) for model validation. This experiment was conducted on the same dairy farm from September to October 2017, but cows were fed another TMR (Appendix A). The outliers were removed according to the mean value ± 3 SD principle and 2259 valid data in DDS + 581 valid data in VDS remained.

All cows were raised in the same barn. The information of feed bunk space and feed barrier design is shown in Appendix A. Feed intake data were collected using automatic weighing troughs (Roughage Intake Control System, Marknesse, The Netherlands). Each feeding station was equipped with an individual recognition system that allowed single cow once to enter a specific feeding bunk and automatically recorded its feeding behavior. All cows were milked three times a day at 0630, 1400, and 2000; fed three times a day at 0700, 1430, and 2030 h ad libitum with 5–10% refusal; and had free access to water. The feed residual was discarded daily before morning feeding. Milk yield was recorded daily, and individual BW was weighed weekly before the morning feeding at the same time; daily BW was calculated using linear interpolation between 2 measurements.

### 2.2. Development and Validation of Forecasting Model

Firstly, the CORR procedure of SAS (version 9.2, SAS Institute, Cary, NC, USA) was used to test the Pearson correlation coefficient between independent variable (DMI-2h, milk yield, BW, DIM, and parity) with total DMI. Then, a stepwise regression analysis was conducted using the PROC REG procedure of SAS to select independent variables that accounted for the highest R^2^ in DMI from the above five variables, and three variables (DMI-2h, milk yield, and BW) were finally selected (see section of Results). These three variables were all continuous variables and have significant linear correlations with DMI. The REG procedure is a simple and general-purpose procedure for both unary and multiple regression analysis [16]. When only a few of independent variables are introduced in the model, and, at the same time, these independent variables are all continuous variables and have significant linear relationships with the dependent variables, REG is suitable and widely used for regression analysis [17,18,19]. Finally, the model to predict DMI was developed using the PROC REG procedure in the SAS software. The prediction model was fitted as follows:Y*_i_* = β_0_ + β_1_X_1*i*_ + β_2_X_2*i*_ + β_3_X_3*i*_ + *e_i_*,
where Y*_i_* is the predicting daily DMI (kg/day) (*i* is the number of observations), β_0_ is the regression intercept, X_1*i*_ is the daily DMI-2h (kg/day), β_1_ is the regression coefficient for DMI-2h, X_2*i*_ is the daily milk yield (kg/day), β_2_ is the regression coefficient for milk yield, X_3*i*_ is the daily BW (kg/day), β_3_ is the regression coefficient for BW, and *e_i_* is the residual error term. Three criteria were calculated to assess and compare the goodness of the models: the R^2^ value of the regression between actual DMI (ADMI) and predicted DMI (PDMI) values; the mean bias between ADMI and PDMI values (ADMI-PDMI); and the root mean square prediction error (RMSPE), calculated as RMSPE = square root [Σ(ADMI-PDMI)2/n], where n is the number of observations. The statistical significance of the ADMI and PDMI was estimated using the paired T-tests module in the GraphPad Prism software (version 8.0), *p* < 0.05 indicated statistical significance.

## 3. Results

The DMI values of dairy cows at different time points in a day are shown in Figure 1. In this study, we observed that cows had the highest DMI values in the first hour after feeding, followed by the second hour after feeding, and the DMI in the following periods was relatively low. Descriptive statistics of DMI and other data of the DDS and VDS in the analyses are provided in Table 1. For individual cows, the total DMI in DDS and VDS ranged 17.3–31.5 and 9.9–35.7 kg/day, respectively. The DMI-2h in DDS and VDS ranged 5.6–27.8 and 1.8–28.9 kg/day, respectively. The ratio of DMI-2h to total DMI ranged 21–100% and 9.0–100% in DDS and VDS, respectively.

The Pearson correlation coefficients of various variables (DMI-2h, milk yield, body weight, DIM, and parity) to total DMI are shown in Table 2. The correlation coefficient of DMI-2h to total DMI was highest (0.467), followed by milk yield (0.376), body weight (BW, 0.330), DIM (−0.266), and parity (0.013). The results of stepwise regression show that R^2^ was highest when all five variables were introduced into the model; however, the contributions of DIM and parity were very limited (Table 3). Therefore, we finally removed these two variables from the model, and the final model was that: DMI (kg/day) = 8.499 + 0.2725 × DMI-2h (kg/day) + 0.2132 × Milk yield (kg/day) + 0.0095 × BW (kg/day).

The evaluation of accuracy and precision of the model are shown in Table 4. The regression relationship between ADMI and PDMI is shown in Figure 2. The R^2^ of our model was medium (0.46) and the mean bias and RMSPE of our model were low (0 and 1.26 kg/day, respectively). In the independent VDS, the R^2^ of our model was higher than the NRC model (0.22 vs. 0.07), and the mean bias and RMSPE were both smaller than NRC model (−0.10 vs. 1.52 kg/day and 1.77 vs. 2.34 kg/day, respectively). We compared the difference between ADMI and PDMI of every day and every cow in VDS, and the results are shown in Figure 3 and Figure 4. For individual day, the first 10 days of ADMI and PDMI are significantly different (*p* < 0.05), while the remaining 20 days are not significantly different (*p* > 0.05) (Figure 3). For individual cows, there are six cows of ADMI and PDMI that are significantly different (*p* < 0.05), while the remaining 16 cows are not significantly different (*p* > 0.05) (Figure 4).

## 4. Discussion

Researchers conducting dairy cow experiments can easily and accurately obtain the feed intake data of dairy cows when the cows are fixed in one position [20,21,22]. However, with the modern construction of dairy farms and the emphasis on the animal welfare, almost all commercial dairy farms nowadays are free-stall barn system, in which all cows are fed in a moving and eating freely cowshed [23]. Under this situation, it is difficult for researchers to measure feed intake data of cows in the experiment [1,23]. Advances in computer technology invented a fully automatic feed intake measuring instrument, which can record feed intake data of cows 24 h a day in time [23,24,25]. However, the equipment is too expensive to be widely applied, and it has to be fixed in one position, which means researchers cannot use it on different farms. Therefore, it is necessary and practical for dairy cow researchers to develop a simple, accurate, and reliable feed intake prediction model.

Firstly, we focused on the daily feeding behavior of dairy cows, which is a factor that has been ignored by many forecasting models developed in the past [4,5]. The daily feeding behavior of dairy cows shows obvious regular changes in this study, that is, the dairy cows eat the majority of ration in the first hour after feeding, and the feed intake within 2 h after feeding accounts for an average of 57% of the total feed intake (ranged from 21% to 100%). The diurnal pattern for feeding behavior agrees with findings of DeVries et al. and Hosseinkhani et al. showing two peaks in DMI [26,27]. These peaks are largely influenced by the time of fresh feed delivery [26]; there are three peaks in this study because of three feed deliveries in one day. At the beginning of the process to develop prediction model of DMI, the feed intake of 1 h after feeding was expected to predict total feed intake, however it had a low correlation with the total feed intake (unpublished), therefore it may not be able to accurately predict total feed intake. Along with the time increases (longer than 2 h), although the correlation increases, it is difficult to record feed intake data for more than 2 h in practice. The correlation coefficient between DMI-2h and total DMI was 0.467 (moderate correlation), which is higher than that between milk yield or BW and DMI, indicating that introducing DMI-2h to predict DMI is feasible and may be more accurate. Milk yield and BW were generally considered to be the variables that are highly correlated with DMI, and almost all prediction models include milk yield (or FCM) and BW (or metabolic weight) to predict DMI [28,29,30]. Moreover, inconsistent with other models, we found that the correlation between DIM or parity and DMI was low, and these two variables had limited improvement in R^2^ of model. This may be attributed to the limited DIM and parity information in our dataset, which only contains DIM in mid and late lactation and parity in 2–4. Therefore, three variables, DMI-2h, milk yield, and BW, were finally chosen to develop the prediction model of DMI in this study. Since the two experimental batches were performed under the same experimental design, with similar dairy cows (DMI, milk yield, DIM, and parity), and the same diet (same ingredients and nutritional level), and we did not observe any batch effects between them by variance analysis, we did not consider random effects. This was also supported by other studies [17,31].

R^2^, mean bias, and RMSPE are the three most commonly used indicators to evaluate model accuracy and precision [32,33]. The higher is the R^2^ value, the better is the model fit and the higher is the explanation of the independent variables (DMI-2h, milk yield, and BW) to the dependent variable (DMI) [34]. The smaller is the mean bias, the better is the accuracy of the model. The smaller is the RMSPE value, the better is the precision of the model [7]. Compared with some reported models [4,5], the mean bias and RMSPE of our model were low (0 and 1.26 kg/day, respectively), which indicated that our model has a better performance to predict daily DMI on accuracy and precision. Meanwhile, we also observed a medium R^2^ value of model (0.46). The medium R^2^ value, which may be because the independent variables and dataset used for model development, is relatively limited in our study. Typically, more independent variables and larger range dataset will result in higher R^2^ value [1], because, with the increase of data, the sum of the squared errors would decrease, and the R^2^ subsequently rise according to the calculation formula of R^2^ (R^2^ = 1−sum of the squared errors/total sum of squares) [34]. The low mean bias and RMSPE values in our model may be the contribution of DMI-2h which was the actual feed intake of cows of over a short time, and it has a strong correlation with total DMI, therefore may reduce the error in predicting daily DMI. Our model has better performance in predicting daily DMI, which is further proved in an independent dataset.

An independent dataset can better evaluate and verify the accuracy and precision of the model [5,33]. Therefore, we used the DMI data from another experiment as a validation data subset to evaluate the model and compared with the NRC model (2001) [6]. The results show that, although the R^2^ value decreased and the mean bias and RMSPE increased, our model had better accuracy and precision compared with the NRC model. Davies et al. reported that RMSPE is a measure of variability of the difference between predicted and reference values for a validation sample dataset; therefore, RMSPE should not be sensitive if the model is robust [35]. The small change of RMSPE in our model between DDS and VDS (1.26 vs. 1.77) indicated a robust performance in DMI prediction. Observing the difference between ADMI and PDMI of every day, we found that the days with significant differences between ADMI and PDMI are mainly displayed in the first nine days (six days showed significant differences). In the study, the dairy cows were not originally raised in the experimental cowshed; they were transferred from another cowshed. It takes some time for cows to adapt to the experimental cowshed and the automatic weighing troughs. Therefore, the dairy cows may be in an adaptation period in the first nine days when feed behavior may be affected, resulting in a significant difference between ADMI and PDMI. In the following 21 days, only four days showed significant difference between ADMI and PDMI, and the difference on Day 21 may be because the ADMI was an abnormal value, although it was not found to be abnormal according to the mean value ± 3 SD principle. Observing the difference between ADMI and PDMI of every cow, we found that most cows were not significantly different. This information shows that our model has a steady performance in predicting daily DMI in this new cow herd. Overall, the model for predicting DMI is credible in the VDS, which indicated that the current model is meaningful and practical and can be used as a reference for scientific researchers who do not have the automatic feed intake recording equipment.

A limitation of this model was that we used few data to develop and verify the model; regression equations developed on a larger dataset would become more robust compared to models developed on a smaller dataset [5]. In addition, as mentioned in the Introduction, feeding behaviors of dairy cows may be different with various feeding management between dairy cow farms. For example, the TMR was delivered three times per day in our experiment, which is commonly used on most commercial dairy farms in China and the USA [36]. However, some dairy farms only deliver TMR once or twice a day in Europe [37]. When TMR are delivered only once a day, the DMI-2h of dairy cows would account for a relatively low proportion of the total DMI (about 10%) [38]. This means that our model may be different in predicting the DMI of cows fed TMR only once a day. Therefore, a larger dataset covering a larger number of animals, various feeding management systems, and different lactation periods is required in the future to verify and improve this model.

## 5. Conclusions

In the current study, we firstly explored the relationship between the DMI-2h and the daily total DMI, and then established a model based on the DMI-2h to predict the total DMI. The equation was: DMI (kg/day) = 8.499 + 0.2725 × DMI-2h (kg/day) + 0.2132 × Milk yield (kg/day) + 0.0095 × BW (kg/day) (R^2^ = 0.46, mean bias = 0 kg/day, and RMSPE = 1.26 kg/day). It was compared with the NRC model and validated in an independent dataset, suggesting that the established prediction model can potentially predict the daily DMI of mid and late lactation dairy cows when fed TMR three times per day in a more convenient way. However, the model has not been validated under different feeding management modes and needs to be evaluated for the prediction performance of DMI for dairy cows only fed TMR once or twice per day.

## Figures and Tables

**Figure 1 animals-11-00104-f001:**
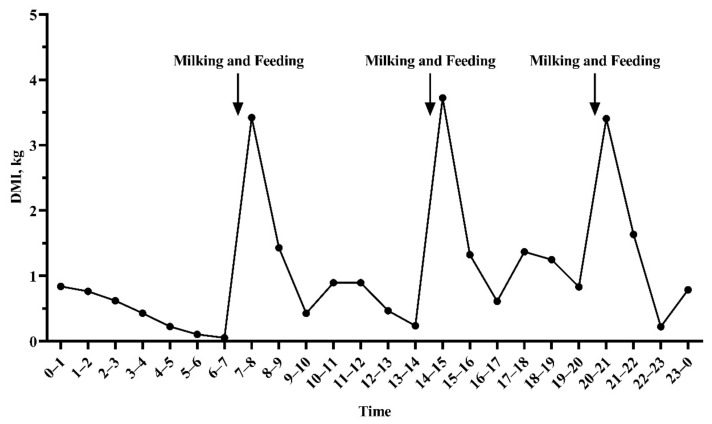
Dry matter intake at various times of the day of dairy cows (development dataset).

**Figure 2 animals-11-00104-f002:**
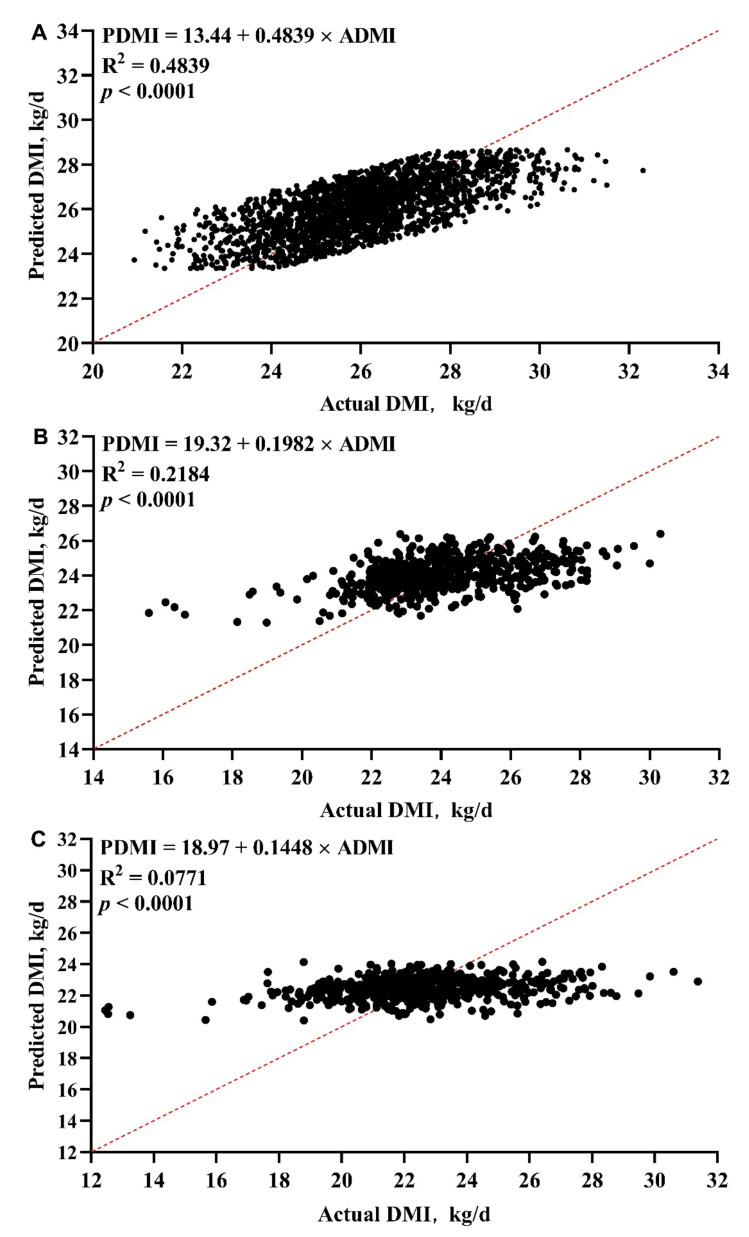
The regression relationship between actual dry matter intake (ADMI) and predicted DMI (PDMI) in development dataset (**A**) and validation dataset (current model (**B**) and NRC model (**C**)).

**Figure 3 animals-11-00104-f003:**
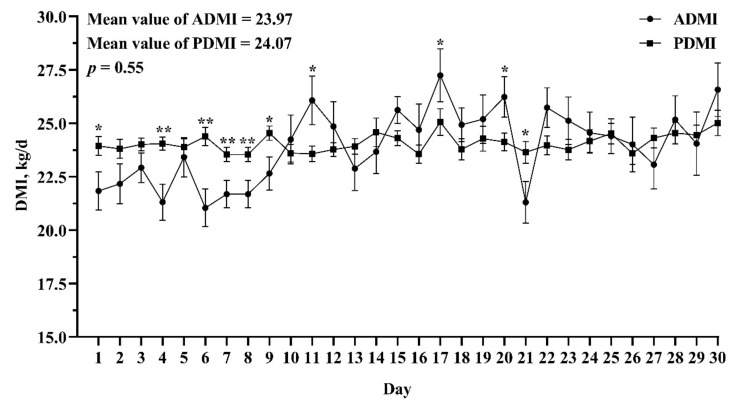
Comparison of actual dry matter intake (ADMI) and predicted dry matter intake (PDMI) in every day in the validation datasets. Bars indicate SEM. The * and ** indicate a significant difference between ADMI and PDMI in the individual day at *p* < 0.05 and *p* < 0.01, respectively.

**Figure 4 animals-11-00104-f004:**
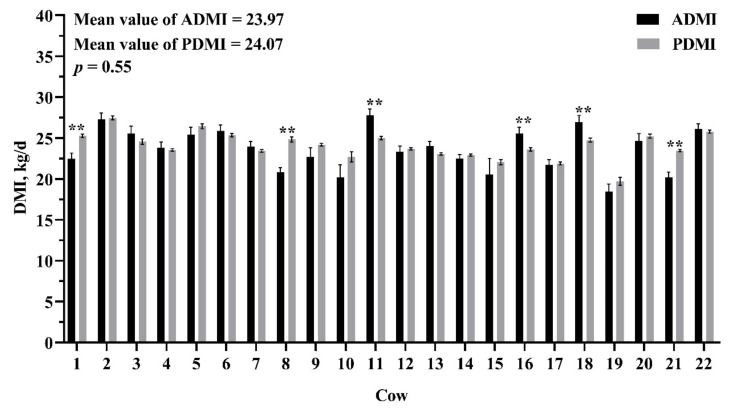
Comparison of actual dry matter intake (ADMI) and predicted dry matter intake (PDMI) in every cow in the validation dataset. Bars indicate SEM. The ** indicate a significant difference between ADMI and PDMI in the individual cow at *p* < 0.01.

**Table 1 animals-11-00104-t001:** Descriptive statistic of all data in development dataset (DDS) and validation dataset (VDS).

Item ^1^	Mean	Minimum	Maximum	SD	CV, %
DDS (*n* = 2259)					
Total DMI, kg/day	26	17.3	31.5	2.77	10.6
DMI-2h, kg/day	14.9	5.6	27.8	3.68	24.6
DMI-2h/Total DMI, %	57.1	21	100	12.83	22.5
Milk yield, kg/day	35	14.4	51.3	5.48	15.7
Body weight, kg	636	465	829	82.6	13
Days in milk	204	114	271	28.5	14
Parity	2.3	2	4	0.57	24.3
VDS (*n* = 581)					
Total DMI, kg/day	23.2	9.9	35.7	4.72	20.3
DMI-2h, kg/day	12.6	1.8	28.9	4.08	32.5
DMI-2h/Total DMI, %	52.4	9	100	15.6	29.8
Milk yield, kg/day	30.4	3.4	48.8	5.85	19.2
Body weight, kg	597	469	732	63.7	10.7
Days in milk	241	127	362	54.5	22.6
Parity	1.6	1	4	0.72	44.3

^1^ DDS, development dataset; DMI, dry matter intake; DMI-2h, dry matter intake of first 2 h after feeding; VDS, validation dataset.

**Table 2 animals-11-00104-t002:** Pearson correlation coefficient of each independent variable to total dry matter intake.

Item ^1^	Total DMI	DMI-2h	MY	BW	DIM	Parity
Total DMI	1					
DMI-2h	0.467	1				
MY	0.376	−0.021	1			
BW	0.33	−0.013	0.047	1		
DIM	−0.266	0.068	−0.32	−0.12	1	
Parity	0.013	0.147	−0.077	0.21	0.017	1

^1^ DMI-2h, dry matter intake of first 2 h after feeding, kg/day; MY, Milk yield, kg/day; BW, Body weight, kg; DIM, Days in milk, day.

**Table 3 animals-11-00104-t003:** R^2^ change with entered more independent variables gradually by stepwise regression.

Entered Variables ^1^	Equation	R^2^	*p*-Value
A	DMI = 22.13 + 0.2628 × A	0.14	<0.01
A, B	DMI = 14.34 + 0.2699 × A + 0.2198 × B	0.37	<0.01
A, B, C	DMI = 8.499 + 0.2725 × A + 0.2131 × B + 0.0095 × C	0.46	<0.01
A, B, C, D	DMI = 11.48 + 0.2776 × A + 0.1995 × B + 0.0091 × C − 0.0108 × D	0.48	<0.01
A, B, C, D, E	DMI = 11.94 + 0.2857 × A + 0.1925 × B + 0.0096 × C − 0.0107 × D − 0.3517 × E	0.48	<0.01

^1^ A, DMI-2h, dry matter intake of first 2 h after feeding, kg/day; B, milk yield, kg/day; C, body weight, kg; D, days in milk, day; E, parity.

**Table 4 animals-11-00104-t004:** Comparison of accuracy and precision between the current model and the NRC model in the development dataset (DDS) and validation dataset (VDS).

	ADMI ^1^	PDMI ^2^	SEM	*p*-Value	Mean Bias	R^2^	RMSPE ^3^
DDS							
Current model	26.04	26.04	0.05	0.99	0	0.46	1.26
VDS							
Current model	23.97	24.07	0.14	0.55	−0.10	0.22	1.77
NRC model (2001) ^4^	23.97	22.44	0.15	<0.01	1.52	0.07	2.34

^1^ ADMI, actual dry matter intake, kg/day; ^2^ PDMI, predicted dry matter intake, kg/day; ^3^ RMSPE, root mean square prediction error, kg/day; ^4^ the equation of NRC model (2001) was: DMI (kg/day) = (0.372 × FCM + 0.0968 × BW^0.75^) × (1 − e (−0.192 × (WOL + 3.67))), where FCM = 4% fat corrected milk (kg/day), BW = body weight (kg), and WOL = week of lactation. The term 1 − e (−0.192 × (WOL + 3.67)) adjusts for depressed DMI during early lactation.

## Data Availability

The data presented in this study are available on request from the authors.

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
