# Peer review of "Predicting Daily Dry Matter Intake Using Feed Intake of First Two Hours after Feeding in Mid and Late Lactation Dairy Cows with Fed Ration Three Times Per Day"

_animals, 2021, doi:10.3390/ani11010104_

Round 1

Reviewer 1 Report

I consider that this paper needs further improvements before to be suitable for publication in Animals. I have two strong concerns. My first concern is about the feeding model used with 3 feeding periods per day. As far as I know, this practice is not very common. 1 feeding time (adding fresh feed material) and one or two moments of brushing food near to the animals is used more at least in Europe. I wonder if this model equating predicting the DMI from data obtained from 3 feeding will work with only 1 feeding. No information about the validation feeding management is provided, and this must be in the Material and Methods section. If the validation data were obtained with different management times, it will prove that the prediction equation will be usefull for other feeding situations. In case that the validation data were from cows fresh feed 3 times a day, this should be clearly stated and the title should refer to this point.

My second concern is about the discrepancy between what is said on the Results lines 124 to 125: “For individual day, only 5 days of ADMI and PDMI are significantly different (P < 0.05), while the remaining 45 days are not significantly different (P > 0.05) (Figure 3)”. As long as it is mentioned in Material and Methods lines 72 to 74: “660 daily DMI data were obtained from 22 Holstein cows (10 primiparity and 12 multiparity, BW = 577 ± 64.3 kg, DIM = 221 ± 54.0 d, cows) in another 30-d experiment to generate the validation dataset (VDS) for model validation.” So, 5 out of 45 days it is not a correct data, 50 days is the duration of the DDS experiment. Also in Figure 3 we can observe that 10 out of 30 days presented significant differences. This means that 1 of every 3 days presented significant differences between ADMI and PDMI. In my opinion, this is not a very promising result, even as indicated in Discussion lines 189 to 190: “Observing the differences between ADMI and PDMI of every day, we found that most of days were not significantly difference.”

Authors must address these concerns in the article before it is eligible for publication.

Author Response

Comments and Suggestions for Authors

My first concern is about the feeding model used with 3 feeding periods per day. As far as I know, this practice is not very common. 1 feeding time (adding fresh feed material) and one or two moments of brushing food near to the animals is used more at least in Europe. I wonder if this model equating predicting the DMI from data obtained from 3 feeding will work with only 1 feeding. No information about the validation feeding management is provided, and this must be in the Material and Methods section. If the validation data were obtained with different management times, it will prove that the prediction equation will be usefull for other feeding situations. In case that the validation data were from cows fresh feed 3 times a day, this should be clearly stated and the title should refer to this point.

AU: Thanks for the reviewer’s valuable comments. This study is designed based on the dairy industry conditions of China, in which the three times feeding is a common practice. To our knowledge, this kind of feeding model is also widely applied in the commercial dairy farms in the North America. The diurnal feeding pattern of cows may be different with different feeding frequency. The reason is when feeding delivery only once a day, the first 2 hours after feeding may not be the peak period of feed intake as observed in our experiment. The feeding frequency and feeding time were consistent between development and validation dataset (VDS). Therefore, we are not able to further evaluate our model in the VDS. Future research is required to investigate the applicability of the model for different feeding frequency. We have revised our title, added the feeding management information of VDS in the Material and Methods section (Lines 100-102, Supplementary Table S1) and supplied related contents in the Discussions section (Lines 234-244) in the revised manuscript.

My second concern is about the discrepancy between what is said on the Results lines 124 to 125: “For individual day, only 5 days of ADMI and PDMI are significantly different (P < 0.05), while the remaining 45 days are not significantly different (P > 0.05) (Figure 3)”. As long as it is mentioned in Material and Methods lines 72 to 74: “660 daily DMI data were obtained from 22 Holstein cows (10 primiparity and 12 multiparity, BW = 577 ± 64.3 kg, DIM = 221 ± 54.0 d, cows) in another 30-d experiment to generate the validation dataset (VDS) for model validation.” So, 5 out of 45 days it is not a correct data, 50 days is the duration of the DDS experiment.

AU: Apologize for the mistake, the manuscript has been corrected.

Also in Figure 3 we can observe that 10 out of 30 days presented significant differences. This means that 1 of every 3 days presented significant differences between ADMI and PDMI. In my opinion, this is not a very promising result, even as indicated in Discussion lines 189 to 190: “Observing the differences between ADMI and PDMI of every day, we found that most of days were not significantly difference.”

AU: Thanks for the reviewer’s comments. As the Figure 3 shows, we found that the days with significant differences between ADMI and PDMI are mainly displayed in the first 9 days (6 days showed significant differences). In the study, the dairy cows were not originally raised in the experimental cowshed, they were transferred from another cowshed. It takes some time for cows to adapt to the experimental cowshed and the automatic weighing troughs. Therefore, we believe that dairy cows may be in an adaptation period in the first 9 days when feed behavior may be affected, resulting in a significant difference between ADMI and PDMI. In the follow 21 days, only 4 days showed a significant difference between ADMI and PDMI. Therefore, we believe that the model for predicting DMI is credible in the VDS. Nevertheless, more data from a larger number of animals, longer time, and different feeding management is required to verify and improve our model. We have added related contents in the Discussion section (Lines 217-226).

Reviewer 2 Report

Dear authors,

Thank you for a neat study on prediction of feed intake in Holstein cows. The data handling was well described and the text easy to follow. It is an appealing approach to include feeding behaviour the first two hours after milking/delivery of fresh feed, in the feed intake prediction model, this time frame is indeed often used in studies of feeding behaviour.

The aim of the study was to establish a prediction model estimating individual daily feed intake from data collected the first couple of hours after milking, to ease the process of data collection of DMI for research purposes. The study suggests that this is achievable by using the prediction model. However, the cows would have to be individually fed in clearly separated feed troughs, individual consumption weighed, and cows would have to be stanchioned for those 2 hours, consequently the suggested method still requires a big effort, and might not fulfil the expressed need “However, it has become much difficult to measure the daily DMI of individual dairy cows that raised in free-stall houses (the major management type in the modern dairy industry)[1]. Therefore, it is necessary to develop an accurate, precise and reliable prediction model of DMI, which will be of practical significance to carry out experimental research on dairy cows.”.

The simple summary could use an English language check.  

Intro

The intro could be expanded with more information on factors influencing feeding behaviour, that would impact the model output, such as feed barrier design, stocking density, feed ration etc.

Please comment on that intake would still have to be recorded with weight individually for two hours.

M&M

Please add info on feed bunk stocking level, and info on the feed barrier design.

Please add which variables were included from start in the stepwise regression.

Results

113 lacks a “the”: all five variables were introduced into the model, but

Clear and descriptive figures.

Discussion

141 Please add references for this statement.

153 lacks an “is”: 0.467 (moderate correlation), which is higher than that between MY or BW and DMI, indicating

Please elaborate on the impact of the study design on the results, beyond the small sample size. Such as feeding behaviour, and factors affecting it.

Conclusion

Conclusion is rather general and lacks details on the study setup that probably impacted model results.

Author Response

Comments and Suggestions for Authors

Dear authors,

Thank you for a neat study on prediction of feed intake in Holstein cows. The data handling was well described and the text easy to follow. It is an appealing approach to include feeding behaviour the first two hours after milking/delivery of fresh feed, in the feed intake prediction model, this time frame is indeed often used in studies of feeding behaviour.

AU: Thanks for the reviewer’s positive comments.

The aim of the study was to establish a prediction model estimating individual daily feed intake from data collected the first couple of hours after milking, to ease the process of data collection of DMI for research purposes. The study suggests that this is achievable by using the prediction model. However, the cows would have to be individually fed in clearly separated feed troughs, individual consumption weighed, and cows would have to be stanchioned for those 2 hours, consequently the suggested method still requires a big effort, and might not fulfil the expressed need “However, it has become much difficult to measure the daily DMI of individual dairy cows that raised in free-stall houses (the major management type in the modern dairy industry) [1]. Therefore, it is necessary to develop an accurate, precise and reliable prediction model of DMI, which will be of practical significance to carry out experimental research on dairy cows.”.

AU: Thanks for the reviewer’s comments. In an experiment, we recorded the feed intake of first two-hour after feeding of 120 cows and found that only two experimenters were required to complete the recording. Firstly, we have observed that almost all cows will be standing and eating when the cows return from milking, at this time, the cow can be easily fixed in feed barrier by the neck clamp. Then, individually cow fed in separated feed troughs and recorded individual consumption weighed. These efforts can be done as easily as recording the feed intake of cows that housed in a tie-stall barn. In most commercial dairy farms, the feed management model is similar, that is, TMR is delivered 2-3 times per day during milking, and the cows can eat fresh feed when they return after milking. And, most dairy farms have similar feed barrier designs (equipped with neck clamps), which can easily fixed cows. Therefore, it feasible for researchers to record feed intake of 2 hours after feeding in many dairy farms.

The simple summary could use an English language check. 

AU: Thanks for the reviewer’s comments, we have revised the summary section according to your suggestion.

The intro could be expanded with more information on factors influencing feeding behaviour, that would impact the model output, such as feed barrier design, stocking density, feed ration etc.

AU: Thanks for the reviewer’s comments, we have revised the introduction section according to your suggestion (Lines 63-77).

Please comment on that intake would still have to be recorded with weight individually for two hours.

AU: The manuscript has been revised as the reviewer suggested (Lines 87-89).

Please add info on feed bunk stocking level, and info on the feed barrier design.

AU: Thanks for your suggestion, we have revised the M&M section according to your comments (Lines 105-106, Supplementary Figure S1 and Figure S2).

Please add which variables were included from start in the stepwise regression.

113 lacks a “the”: all five variables were introduced into the model, but

Clear and descriptive figures

AU: The manuscript has been revised as the reviewer suggested.

141 Please add references for this statement.

AU: Thanks. we have added references to support this statement.

153 lacks an “is”: 0.467 (moderate correlation), which is higher than that between MY or BW and DMI, indicating

AU: The manuscript has been revised as the reviewer suggested.

Please elaborate on the impact of the study design on the results, beyond the small sample size. Such as feeding behaviour, and factors affecting it.

AU: Thanks for your comments, the manuscript has been revised it based on your suggestion (Lines 234-244).

Conclusion is rather general and lacks details on the study setup that probably impacted model results.

AU: Thanks for the reviewer’s comments, we have revised it according to your suggestion (Lines 250-252).

Round 2

Reviewer 1 Report

Authors addressed the concerns mentioned in the first round of revision and I consider that the paper is now suitable for publication.

Author Response

Thanks for the reviewer's positive comments.